# Vitamin D Status and Steatohepatitis in Obese Diabetic and Non-Diabetic Patients

**DOI:** 10.3390/jcm11185482

**Published:** 2022-09-18

**Authors:** Laura Taban, Dana Stoian, Bogdan Timar, Daniela Amzar, Calin Adela, Alexandru Motofelea, Andreea Borlea, Romain Frisoni, Nadege Laguerre

**Affiliations:** 1PhD School Department, “Victor Babes” University of Medicine and Pharmacy Timisoara, 2nd Eftimie Murgu Square, 300041 Timisoara, Romania; 2Second Department of Internal Medicine, “Victor Babes” University of Medicine and Pharmacy Timisoara, 2nd Eftimie Murgu Square, 300041 Timisoara, Romania; 3Service de Chirurgie Digestive, Cancérologique, Bariatrique et d’urgence de Mercy Centre Regional Hospitaliere Metz-Thionville, Hopital Mercy, 57530 Ars-Laquenexy, France; 4Internal Medicine, Department of Endocrinology, Centre Regional Hospitaliere Metz-Thionville, Hopital Mercy, 57530 Ars-Laquenexy, France

**Keywords:** steatohepatitis, NASH, diabetes, obesity, vitamin D

## Abstract

Background and Aims: The presence of steatohepatitis in obese patients can be multifactorial. The current study tries to determine the differences between diabetic and non-diabetic patients regarding the presence of steatohepatitis. We evaluated sequential liver samples and collected the times of bariatric surgery to assess the presence of NASH in patients with obesity, in the circuit of bariatric surgery. Methods: We performed a retrospective study of 49 patients presenting high-grade obesity in the circuit of bariatric surgery, with liver biopsy. The patients underwent bariatric surgery at a single center in France and were followed for 2 years. The liver biopsies were performed intraoperatively on all 49 patients before the bariatric surgery. The primary endpoint of the study was to evaluate the relationships between steatohepatitis/liver fibrosis and the presence of diabetes and to evaluate the current relationships between the biochemical work-ups. Special importance was accorded to the correlations between vitamin D levels and the presence of hepatic steatosis, due to the antifibrogenic pattern in the liver, as shown in many important papers in the field. Results: Significant correlations were found between the presence of liver fibrosis and the presence of diabetes (*p* = 0.022), but not regarding the antidiabetic treatment. An important correlation was found between the vitamin D levels and the presence of liver fibrosis, as well as with the levels of A1C hemoglobin and LDL cholesterol levels. Conclusions: Vitamin D deficiency presents a strong correlation with hepatic steatosis in individuals with morbid obesity. Correcting vitamin D deficiency may present a beneficial role in treating hepatic steatosis, diabetes, and cardiovascular risk in patients with morbid obesity.

## 1. Introduction

Current epidemiological data present a worldwide epidemic of obesity and type 2 diabetes mellitus (T2DM), with an emerging prevalence of non-alcoholic fatty liver disease or metabolic (dysfunction) associated fatty liver disease (MAFLD) among these patients [1].

Metabolic (dysfunction)-associated fatty liver disease (MAFLD) is the principal cause of abnormal liver function biochemical evaluation and liver disease among obese adults [2,3]. Metabolic (dysfunction)-associated fatty liver disease (MAFLD) involves increased concentrations of triglycerides in hepatocytes, determining an increase in liver mass by 5–10% [4] in subjects with low or no alcohol consumption. MAFLD can present different degrees of histological modifications, from simple hepatic steatosis and non-alcoholic steatohepatitis (NASH) to an inflammatory reaction and cellular damage in the liver such as the ballooning of hepatic cells, or the presence of different degrees of fibrosis, progressing to cirrhosis and eventually to hepatocellular carcinoma. MAFLD is commonly associated with obesity, type 2 diabetes (T2DM), dyslipidemia, and insulin resistance, all components of the metabolic syndrome, as well as the presence of metabolic dysregulation and colonic diverticulosis [5], implying that MAFLD is the hepatic involvement of the metabolic syndrome [6].

Dyslipidemia and insulin resistance (IR) present an association with excessive triglyceride deposits in the liver hepatocytes [7,8]. This accumulation of increased gluconeogenesis decreases glycogen synthesis, exacerbates insulin resistance, and increases hepatic inflammation, producing a progression of liver disease from healthy to fibrosis, to cirrhosis, and therefore producing a higher risk of hepatocarcinoma [5,7,8]. Weight loss is a very important and decisive change in controlling MAFLD. Bariatric surgery on MAFLD has presented beneficial effects, improving histological results, as stated in some studies, with a resolution of NASH in 75% of cases, a further reduction in fibrosis in 35% of patients [9,10], and a partial reversion of cirrhosis; these results were not achieved with other treatment, such as diet and habit changing [10]. In 2017, the International Diabetes Federation published data estimating the rising prevalence of type 2 diabetes (T2DM) in 415 million humans and with an expectation of further growth to 642 million by 2040 [11]. Given this situation, better control of obesity should be achieved by better educating people and anticipating these facts by controlling weight loss via bariatric surgery and metabolic surgery, further improving global outcomes, and controlling cardiovascular risks, thus decreasing the economic burden of what is to come [12].

The prevalence of MAFLD has been reported to be in the 15–30% range in the general population in various countries [13] and increasing. Compared with non-diabetic patients, people presenting type 2 diabetes appear to have an increased risk of MAFLD appearing and surely present a higher risk of developing fibrosis and cirrhosis [2,6].

Vitamin D is an important hormone that presents multiple roles apart from the homeostasis of bone. Important papers in the field have presented a crucial role of vitamin D in the modulation of the immune system, presenting important anti-inflammatory and antifibrogenic roles, especially in liver fibrosis [14,15,16]. A recent meta-analysis has depicted an important association between vitamin D status and the presence of MAFDL, both in adults and children [16,17,18,19].

The main objective of this study is to associate the histology results of liver biopsy in patients who underwent bariatric surgery for obesity with the presence of diabetes and the status of vitamin D. The result is to estimate the prevalence of NASH and fibrosis in this group of patients and to verify if there is any correlation between the biochemical features, such as vitamin D, lipids, biochemical profile, and the current medications.

## 2. Materials and Methods

In this retrospective study, 49 patients with severe obesity, diabetic and non-diabetic patients who underwent bariatric surgery by the same team from the Mercy Hospital, Metz from 2020 to 2021, underwent perioperative liver biopsy.

Patients with reported alcohol consumption greater than 140 g/week for men and 70 g/week for women [7], current use or ingestion of hepatotoxic treatments, current other liver diseases such as viral hepatitis or liver carcinoma, or the previous bariatric surgery of intragastric balloon insertion, were excluded from the cohort.

For the cohort, anthropometric data before surgery were recorded, including body mass index (BMI), biochemical work-up, and histopathological results. The cohort consisted of patients without diabetes, patients with T2DM in treatment with insulin-basal or rapid analogs or other oral antidiabetic drugs in the preoperative assessment, and patients without active diabetes but with glycosylated hemoglobin (HbA1c) >6.5. The study was approved by the Local Ethics Committee (II/11.2020) and the study was following the Ethics Code of the World Medical Association (Declaration of Helsinki, Seoul, Korea, October 2008).

Regarding the histopathological results, we used the MAFLD activity score (NAS) system, created by Kleiner et al. [20], to assess the degree of hepatic steatosis, the presence of ballooning, inflammation activity, and fibrosis degree.

The NAS system [21] evaluation comprises the scores for histological evaluation (0–8), the extent of steatosis (0–3), lobular inflammation (0–3), and ballooning (0–2). To quantify the liver steatosis, the following were considered:Less than 5% of steatosis received a score of 0;A total of 5% to 33% received a score of 1;An amount of 34% to 66% was noted as score of 2;Greater than 66% was noted as score of 3.

For the correct diagnosis of NASH [22], steatosis must be associated with hepatocyte ballooning and/or inflammatory infiltration. NASH activity was graded from 0 to 2, according to the presence of ballooning as none (0), mild (1), or severe (2). Lobular inflammation was noted as the absence of foci (0), the presence of 1–2 foci in twenty fields (1), the presence of 3–4 foci in twenty fields (2), and the presence of more than 4 foci in twenty fields (3).

We presented the degree of fibrosis as:Stage 0, without presence of fibrosis;Grade 1a, when mild fibrosis was found in zone 3;Grade 1b, when moderate fibrosis was found in zone 3;Grade 1c, when only periportal/portal fibrosis was found;Grade 2, when periportal/portal fibrosis and zone 3 were present;Grade 3, when bridging fibrosis was identified;Grade 4, when cirrhosis was found.

An additional elastographic measurement was added, and the degree of hepatic fibrosis was measured with transient elastography with ARFI-Fibroscan. Elastography is a validated complementary method to 2B ultrasound evaluation [23,24,25]. It is simple, non-invasive, and operator-independent and with a good correlation with the histopathological liver biopsy results [26].

The biochemical work-up was assessed preoperatively, and it included a complete lipid profile—total cholesterol, HDL cholesterol, LDL cholesterol, triglycerides, a complete blood count with platelets count, the liver enzymes with total bilirubin level (BT), glucose, and hemoglobin A1C.

### Statistical Analysis

Data were collected and analyzed using SPSS version 26 (SPSS Inc., Chicago, IL, USA) and are presented as means ± standard deviation for continuous variables with Gaussian distribution, and as median and interquartile range for continuous variables without Gaussian distribution or percentages for categorical variables. To assess the significance of the differences studied between groups, we used the Student’s *t*-test, the analysis of variance (means and Gaussian populations), the Mann–Whitney U test, the Kruskal–Wallis test (medians and non-Gaussian populations), or the χ2 (proportions) test. Continuous-variable distributions were tested for normality using the Shapiro–Wilk test and for equality of variance using Levene’s test. The strength of association between two continuous variables from non-Gaussian populations was evaluated using Spearman’s correlation coefficient. Sample-size calculation was performed before the study, aiming to provide a confidence level of 95%. In this study, *p*,0.05 was considered the threshold for statistical significance.

## 3. Results

Our study included 49 patients, 18 diabetics (2 patients with type 1 diabetes and 16 patients with type 2 diabetes mellitus) and 31 non-diabetic patients, who underwent bariatric surgery by the same operating team. Of those patients, 14 (28.6%) had grade II obesity and 35 (71.4%) had morbid obesity. From the cohort, 34 patients were women and 15 were men, with ages between 29 and 63 years, and the median age was 50 years.

Liver biopsy was performed intraoperatively and liver fibrosis in the patients was evaluated preoperatively by using transient elastography.

Ten patients did not present liver fibrosis by elastographic evaluation, confirmed by the histopathological evaluation of the liver biopsy. According to the liver histological findings, 7 patients (14.3%) had cirrhosis, 3 patients (6.1%) had F3 fibrosis, 8 patients (42.0%) had F2 fibrosis, and 21 patients (20.44%) had F1 fibrosis, followed by 10 patients with no fibrosis (36.4%).

Thirty-one patients presented type 2 diabetes with oral medication and one patient presented type 1 diabetes with insulin treatment. When analyzing their current antidiabetic treatment, we found that 27 (44.9%) had a treatment with Metformine, 6 (9.1%) with dipeptidyl peptidase-4 inhibitor (DPP4), and 3 (6.1%) with sulphamides, followed by 2 (4.1%) in treatment with sodium-glucose cotransporter-2 (SGLT2) inhibitors.

The baseline characteristics of the studied cohort are presented in Table 1. The HbA1C results presented in Table 1 are from diabetic and non-diabetic patients, as all of them were evaluated with this parameter before surgery.

There were significant differences regarding the presence of liver fibrosis between the patients with diabetes and the non-diabetic cohort *p* = 0.022 (Chi-square) (Figure 1).

Significant correlations were found when regarding the biochemical workups and the presence of fibrosis.

In Table 2 are presented the clinical correlations of the histopathological findings, gender, body mass index, and the presence of diabetes with the total preoperative vitamin D levels.

The correlation between vitamin D and biochemical work-ups was evaluated using Pearson’s correlation coefficient r. The statistical relationships between vitamin D and fibrosis, LDL, HDL triglycerides, and Hba1c was evaluated.

The results (R coefficient, 95% confidence intervals for each, and a *p*-value for the difference between the r values) were as follows: low-density lipoprotein (LDL) cholesterol and vitamin D correlation (R 0.39, 95% CI [0.12–0.6], *p* = 0.007), high-density protein (HDL) cholesterol and vitamin D correlation (R −0.38, 95% CI [−0.6–−0.11], *p* = 0.025), triglycerides and vitamin D concentration (R −0.33, 95% CI [−0.56–−0.061], *p* = 0.018), hemoglobin A1c and vitamin D (R 0.38, 95% CI [0.11–0.6], *p* = 0.006), and liver fibrosis and vitamin D concentration (R 0.41, 95% CI [0.15–0.62], *p* = 0.003). We observed moderate positive correlation between vitamin D and LDL cholesterol, as well as a positive correlation between liver fibrosis and vitamin D and a moderate negative correlation between HDL cholesterol and triglycerides.

We observe a positive correlation between the mean vitamin D value and fibrosis (r = 0.41, *p* = 0.003) and between the Hba1c and vitamin D concentrations (r = 0.38, *p* = 0.006), as presented in the scatterplot in Figure 2, indicating a positive linear association between vitamin D, fibrosis, and hemoglobin A1c.

## 4. Discussion

In this study, we performed a retrospective analysis in patients with morbid obesity who underwent bariatric surgery. Our primary endpoint was to evaluate the differences between diabetic and non-diabetic patients regarding the presence of steatohepatitis. We did find significant differences regarding the presence of diabetes in this cohort of patients, as it is a plausible difference considering the presence of the metabolic syndrome in patients with type 2 diabetes. However, when assessing their preoperative biochemical status, we also observed a significant correlation between the vitamin D status and the presence of liver fibrosis.

Literature studies have identified a strong correlation between obesity, liver steatosis, and low vitamin D levels.

Numerous literature studies have determined the association between vitamin D and MAFLD. Targher et al. [27] conducted the first study proving the association and confirming that vitamin D levels were lower in patients presenting with liver steatosis compared to controls. Furthermore, vitamin D serum concentrations can predict the histological severity of MAFLD, and NASH patients have lower vitamin D levels compared to the levels found in patients with isolated fatty liver disease, proven by many other literature studies [18,28,29,30]. Knowing that vitamin D undergoes an important biological activation in the liver, it is biologically possible that chronic liver diseases, including MAFLD, could alter the vitamin D metabolism, decreasing its plasmatic levels [31]. However, further research is necessary.

It was observed in many studies including meta-analysis that vitamin D concentrations are linked to a decrease in the risk of occurrence of the metabolic syndrome, diabetes, and cardiovascular diseases [32,33,34].

Furthermore, strong evidence from the literature database shows that MAFLD can present serious extrahepatic manifestations such as cardiovascular disease, polycystic ovarian syndrome, chronic kidney disease, and hypothyroidism, as well as a higher risk of developing various infections [16,35].

In addition, there is a significant correlation between vitamin D status and A1c hemoglobin, suggesting that a deficiency in vitamin D could lead to worse glycemic control independently from the antidiabetic treatment.

Numerous publications propose that low levels of vitamin D are strongly associated with features of the metabolic syndrome [28,31], suggesting that a normalization in the serum concentrations of vitamin D could impact the total outcome in the cardiovascular risk in patients presenting with type 2 diabetes [31]. A recent study suggested that there is a high prevalence of hypovitaminosis D among patients with type 2 diabetes, especially in those associating obesity and patients with dyslipidemia with poor glycemic control and in those with longer diabetes durations, with a higher prevalence in women [36].

A meta-analysis concluded that vitamin D levels above 25 ng/mL were associated with a 43% lower risk of type 2 diabetes compared to levels <14 ng/mL, thus presenting that vitamin D supplementation can improve insulin resistance and the baseline glucose tolerance in this group of patients [31,37]. Similarly, another meta-analysis concluded that vitamin D supplementation can improve insulin resistance but has a weak effect [38].

We also found a significant positive correlation between vitamin D and LDL cholesterol levels, suggesting that a deficiency in vitamin D can have an important impact on the cardiovascular impact and re-evaluating the management of patients at risk.

A recent meta-analysis reviewed 41 randomized trials comprising 3434 patients, mostly female patients, comparing the changes from baseline vitamin D levels and cholesterol levels to follow-up vitamin D supplementation, concluding that vitamin D supplementation appears to have a beneficial effect on reducing serum total cholesterol, LDL cholesterol, and triglyceride levels but not HDL cholesterol levels. A deficiency in vitamin D can increase the risk of cardiovascular diseases [39].

There are some limitations to our work that must be acknowledged. Firstly, this is a retrospective study and therefore we could not perform any methods directed at the diagnosis of non-alcoholic fatty liver disease. We had a relatively small cohort of patients, with a slightly higher number of diabetic patients. Furthermore, we only evaluated the parameters in one moment; we did not follow the patients’ parameters after the bariatric surgery to compare the values before and after surgery, as values can vary according to acute events and offending factors.

## 5. Conclusions

Taking all the results into account, vitamin D deficiency seems to be associated with a higher risk of hepatic steatosis in patients with obesity. Moreover, vitamin D seems to present an important role in glycemic control in patients with type 2 diabetes and in the cardiovascular risk, presenting a direct correlation with the levels of LDL cholesterol. More studies, preferably prospective and randomized on larger cohorts, are necessary to establish the direct relationship between vitamin D with liver steatosis and the potential role of vitamin D supplementation in the management of this disease.

## Figures and Tables

**Figure 1 jcm-11-05482-f001:**
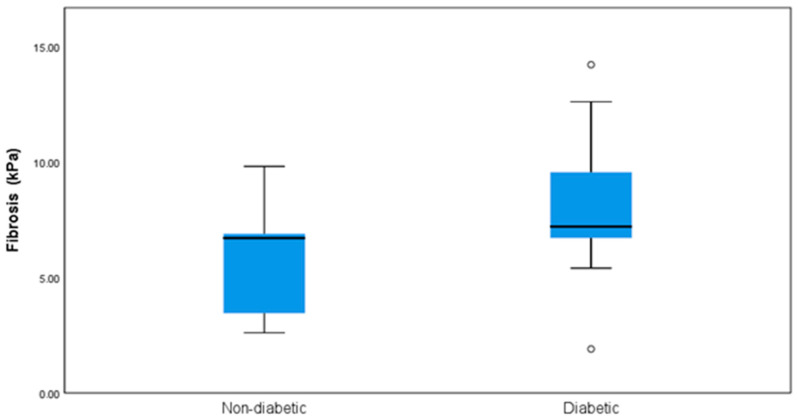
Statistical differences between non-diabetic and diabetic patients regarding the presence of liver fibrosis.

**Figure 2 jcm-11-05482-f002:**
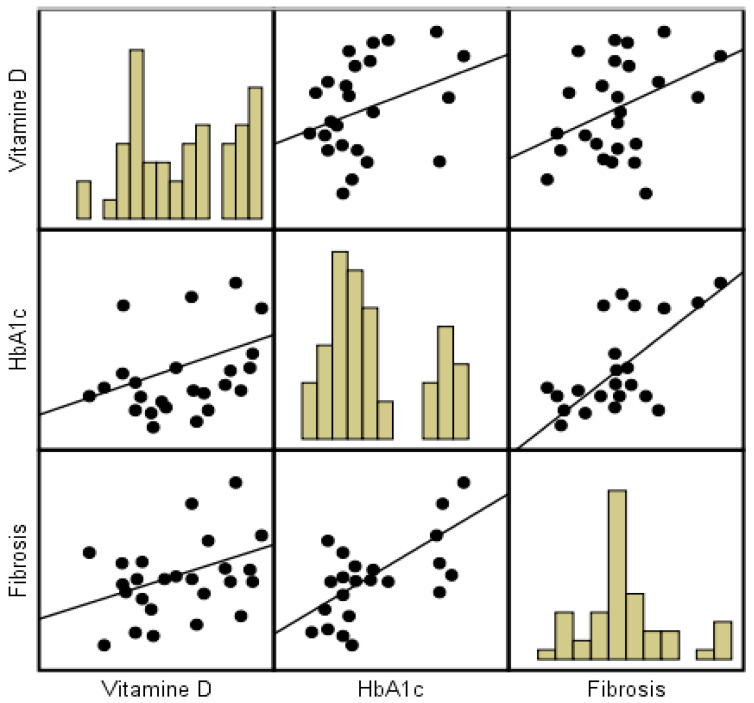
Correlation between vitamin D (ng/dL), Hba1c, and fibrosis.

**Table 1 jcm-11-05482-t001:** Patient baseline characteristics.

Parameter	Value
Age (years) ^b^	50 (20)
BMI ^b^	43.20 kg/m^2^ (6)
Total bilirubin ^a^	5.3 mg/dL (2.6)
Albumin ^a^	40.4 g/L (4.6)
Platelets ^a^	269,061 cc/m^3^ (57,931)
Vitamine D ^a^	23.9 ng/mL (9.5)
Glucose ^b^	1.05 g/L (0.38)
HbA1c ^b^	6.1% (1.3)
HDL cholesterol ^b^	0.44 g/L (0.23)
Triglycerides ^b^	1.36 g/L (1.2)
AST ^b^	22 UI/L (12)
ALT ^b^	26 UI/L (19)
GGT ^b^	26 UI/L (52)
Fibroscan ^a^	7.33 kPa (2.95)

**Notes:** ^a^ Numeric variables with Gaussian distribution. Results are presented as mean ± standard deviation. ^b^ Numeric variables without Gaussian distribution. Results presented as median and interquartile range. **Legend:** BMI—body mass index; HbA1c—hemoglobin A1c; HDL cholesterol—high-density lipoprotein cholesterol; LDL cholesterol—low-density lipoprotein cholesterol; AST—aspartate transaminase; ALT—alanine transaminase; GGT—gamma-glutamyltransferase.

**Table 2 jcm-11-05482-t002:** Vitamin D association between groups.

Parameters		N	Vitamin D (ng/dl)	*p* Value
Fibrosis	0	10 (36.4%)	19.7 (7.2)	0.001
	1	21 (20.44%)	24.1 (6.8)	
	2	8 (42.29%)	20.5 (12.7)	
	3	3 (6.1%)	39.9	
	4	7 (14.3%)	34.7 (3.8)	
Steatosis	0	4 (12.1%)		0.053
	1	14 (28.6%)	20.8 (7.4)	
	2	22 (44.9%)	20.5 (12.7)	
	3	9 (18.4%)	28.2 (10.5)	
	4	4 (8.2%)	35	
NASH	0	7 (14.3%)	20.2 (7.4)	0.094
	1	35 (71.4%)	24.6 (9.3)	
	2	3 (6.1%)	28.8 (16.2)	
	3	4 (8.2%)	35	
Ballooning	0	12 (24.5%)	24.7 (11.3)	0.308
	1	18 (63.3%)	24.2 (9.1)	
	2	6 (12.2%)	30.7 (7.3)	
Lobular inflammation	0	11 (22.4%)	26.5 (10.5)	0.071
	1	25 (51%)	22,1 (8,9)	
	2	13 (26.5%)	29.7 (8.6)	
Gender	M	15 (30.6%)	25.4 (10.5)	0.103
	F	34 (69.4%)	25 (9.3)	
BMI scale	Grade II obesity	14 (28.6%)	30.3 (9.4)	0.098
	Morbid obesity	35 (71.4%)	23 (9)	
Diabetes	Non-diabetic	17 (34.7%)	22 (7.2)	0.093
	Type 2 diabetes	32 (65.3%)	26.8 (10.4)	

**Notes:** Categorical variables are expressed as count and frequency. Results are presented as mean and SD. *p* values are from ANOVA and independent T-test.

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
