# Peer review of "Vitamin D Status and Steatohepatitis in Obese Diabetic and Non-Diabetic Patients"

_jcm, 2022, doi:10.3390/jcm11185482_

Round 1
Reviewer 1 Report
Manuscript ID: JCM-1892692I request the authors to meticulously correct and revise the manuscript so that the medical community and readers can clearly understand the work done by the authors.
Below are my suggestions:
The abstract says: The current study tries to determine the differences between diabetic and non-diabetic patients regarding the presence of steatohepatitis
Title Says: Vit D and steatohepatitis in obese diabetic and non - diabetic, is there any difference?
It should be either diabetic and non-diabetic or obese diabetic and non - diabetic.
If possible, authors can modify the title after reflecting on their work.
Introduction part of this manuscript should consist of the impact of Vit D levels or the reasons behind the analysis of Vit D in patients or medical conditions included in this study.
Should improve the sentences regarding NAFLD in the introduction part since it was ambiguous.
Language correction can make the manuscript more clearer for the readers.
Also, needs proper and careful formatting.
Can avoid the repetition of the same word continuously in the consecutive lines, it will elevate the legibility of your article.
Though the authors are trying to present the information about NAFLD, in many places it is unclear to articulate what they are up to.
Line No: 71 - It should be "liver" biopsy
For the cohort, authors have included diabetic and non-diabetic, what about obese diabetic?
It was given, 49 patients with severe obesity were included in this study. But the results of this displays only about non diabetic and diabetic patients (Fig 1).
Many errors in the usage of scientific terminologies. Please correct them.
Please clarify, what do you mean by severe obesity
Table 3: Very ambiguous, check the whole table and most importantly results again, please present the data clearly.
Line Numbers: 165 to 169 - recheck the facts.
Please stick with the key factor regarding NASH.
Authors have used different terms such as morbid obesity, diabetic or non-diabetic, obese, bariatric or bariatric with diabetes, morbid underwent bariatric surgery....
In Table 2: it should be "NASH"
Table 1: Authors should mention the parameters very carefully and correctly
Author Response
Honorable reviewer,
Thank you for your time and review suggestions, we found them very helpful. We have made the changes according to your suggestions, as it follow:
The abstract says: The current study tries to determine the differences between diabetic and non-diabetic patients regarding the presence of steatohepatitis
Thank you for you suggestion. We have rewritten the abstract and corrected the material and methods, to clearly state the purpose of this study.
“ The primary endpoint of the study was to study the relationships of the steatohepatitis/liver fibrosis with the presence of diabetes and antidiabetic drugs and to evaluate if there are any current relationships between the biochemical workups. A special importance was accorded to the correlations between vitamin D levels and the presence of hepatic steatosis, due to the antifibrogenic pattern in the liver, as shown in many important papers in the field.”
- Title Says: Vit D and steatohepatitis in obese diabetic and non - diabetic, is there any difference?
Thank you, we have now corrected the tittle according to review suggestions to: “ Vitamin D status and steatohepatitis in obese diabetic and non-diabetic patients”
- It should be either diabetic and non-diabetic or obese diabetic and non - diabetic.
Thank you, we would like to specify that all patients in the current study were obese, as the patients were in bariatric surgery unit. We have further divided them into diabetic and non-diabetic patients.
If possible, authors can modify the title after reflecting on their work
Thank you, we have now corrected the tittle according to review suggestions to: “ Vitamin D status and steatohepatitis in obese diabetic and non-diabetic patients”
Introduction part of this manuscript should consist of the impact of Vit D levels or the reasons behind the analysis of Vit D in patients or medical conditions included in this study.
Thank you, we have introduced a pertinent paragraph explaining the motivation of the study based on the current literature findings.
“Vitamin D is an important hormone that presents multiple roles apart from the homeostasis of bone. Important papers in the field have presented a crucial role of vitamin D in the modulation of the immune system, presenting important anti-inflammatory and anti-fibrogenic roles, especially in liver fibrosis [14–16]. Recent meta-analysis depicts an important association between vitamin D status and the presence of MAFDL, both in adults and children [17–20].
The main objective of this study is to associate the histology results of liver biopsy in patients who underwent bariatric surgery for obesity, with the presence of diabetes and the status of vitamin D. The result is to estimate the prevalence of NASH and fibrosis and to verify if there are any correlation with the biochemical features such as vitamin D, lipids biochemical profile and the current medications.”
Should improve the sentences regarding NAFLD in the introduction part since it was ambiguous.
Thank you ,we have made pertinent changes, specifically clarifying NAFLD, changing to the current denomination of the disease - metabolic (dysfunction) associated fatty liver disease.
7. Language correction can make the manuscript more clearer for the readers.
Thank you, we have reviewed our English, as we are not native English speakers.
8. Also, needs proper and careful formatting.
The formatting has been changed according to the demands of the journal.
9. Can avoid the repetition of the same word continuously in the consecutive lines, it will elevate the legibility of your article.
Thank you, we have reviewed the discussion part, where, as you mentioned we had the same word in consecutive lines.
10. Though the authors are trying to present the information about NAFLD, in many places it is unclear to articulate what they are up to.
Thank you, we have changed this fact and clearly stated our aim from the abstract and continuing in the introduction and material and methods.
- Line No: 71 - It should be "liver" biopsy
Thank you, we have corrected the error.
12. For the cohort, authors have included diabetic and non-diabetic, what about obese diabetic?
It was given, 49 patients with severe obesity were included in this study. But the results of this displays only about non diabetic and diabetic patients (Fig 1).
Thank you, all patients included in the study are obese patients, as they have been recruited from the bariatric ward. The results in figure 1 – represent the differences between obese diabetic and obese non diabetic patients.
- Many errors in the usage of scientific terminologies. Please correct them.
Thank you, we have corrected our terminology errors.
14. Please clarify, what do you mean by severe obesity
Thank you, in the current study, we included patients with BMI> 35 ( grade II obesity). We have corrected this point.
15. Table 3: Very ambiguous, check the whole table and most importantly results again, please present the data clearly.
Thank you for your suggestion, we have cut out Table 3 and introduced a more detailed paragraph with our findings.
“ The results ( R coefficient, 95% confidence intervals for each and a p‐value for the difference between the r values) were as it follows: low-density lipoprotein (LDL) cho-lesterol and vitamin D correlation ( R 0.39, 95% CI [0.12-0.6], p 0.007), High-density protein (HDL) cholesterol and vitamin D correlation ( R -0.38, 95% CI [-0.6- - 0.11], p 0.025), triglycerides and vitamin D concentration ( R -0.33, 95% CI [-0.56 - - 0.061], p 0.018), hemoglobin A1c and vitamin D ( R 0.38, 95% CI [0.11-0.6], p 0.006), liver fibrosis and vitamin D concentration ( R 0.41, 95% CI [0.15-0.62], p 0.003). We can observe a moderate positive correlation between Vit D with LDL and Fibrosis and moderate negative correlation between HDL and Tri-glycerides.”
16. Authors have used different terms such as morbid obesity, diabetic or non-diabetic, obese, bariatric or bariatric with diabetes, morbid underwent bariatric surgery....
Thank you for your suggestions, we have corrected and clarified the terms, accordingly.
17. In Table 2: it should be "NASH"\
Thank you, we have corrected this error.
18. Table 1: Authors should mention the parameters very carefully and correctly
Thank you for your suggestion, we have corrected the parameters and added a legend to table 1, to clarify the confusions.
Table 1. Patient baseline characteristics.
|
Parameter |
Value |
|
Age(years)b |
50 (20) |
|
BMIb |
43,20 kg/m2(6) |
|
Total Bilirubina |
5,3 mg/dl(2,6) |
|
Albumina |
40,4g/l (4,6) |
|
Plateletsa |
269061cc/m3(57931) |
|
Vitamine D a |
23,9 ng/ml (9,5) |
|
Glucoseb |
1,05 g/l (0,38) |
|
HbA1cb |
6,1% (1,3) |
|
HDL cholesterolb |
0,44 g/l (0,23) |
|
Triglyceridesb |
1,36 g/l (1,2) |
|
ASTb |
22 UI/L (12) |
|
ALTb |
26 UI/L (19) |
|
GGTb |
26 UI/L (52) |
|
Fibroscana |
7,33 kPa (2,95) |
Notes: a Numeric variables with Gaussian distribution. Results presented as mean ± standard deviation. b Numeric variables without Gaussian distribution. Results presented as median and interquartile range.
Legend: BMI – body mass index, HbA1c – hemoglobin A1c, HDL cholesterol – high density protein cholesterol, LDL cholesterol – low-density protein cholesterol, AST - Aspartate transaminase, ALT- Alanine transaminase, GGT - Gamma-glutamyltransferase.
We hope that the changes are according to your recommendations.
Thank you,
The authors
Reviewer 2 Report
Dear Authors,
Congratulations on your research idea and well done work. As you mentioned, the manuscript has limitations in small group of patients. I understand that bariatric surgery is not the surgery which is done several times a day as e.g. cholecystectomy.
My comments:
1. The title has to be changed. Should be : Vitamin D levels and steatohepatitis....
2. Please standardize the spelling of NAFLD (nonalcoholic or non-alcoholic).
3. Because in the title is vitamin D I suggest to give few sentences about problem of vitamin D difficiency in NAFLD to explain the aim of the study.
4. The main objective of this study is to associate the histology results of liver biopsiesin patients.........(lines 71-74). In the results there is no answer to main objective of the study. Please re-edit the aim of the study.
5. There is no information about the source of grading system used in determination of fibrosis grade [refferences].
6. Was there a difference in the grade of fibrosis between grade II obesity and moprbid obesity patients?
7. Lines 165-169 (Literature studies............cardiovascular diseases [18]) should be placed in the discussion not in the results.
8. Were there any patients in the study group with very low vitamin D concentrations, e.g. <10 or 14 ng / mL, and did they have a higher degree of fibrosis or steatosis? If possible, it would be worth checking. You mentioned about such situation in the discussion.
Author Response
Honorable reviewer,
Thank you for your time and review suggestions, we found them very helpful. We have made the changes according to your suggestions, as it follow:
- The title has to be changed. Should be : Vitamin D levels and steatohepatitis....
Thank you for your suggestion, we have changed the title to “Vitamin D status and steatohepatitis in obese diabetic and non-diabetic patients
- Please standardize the spelling of NAFLD (nonalcoholic or non-alcoholic)
Thank you, we corrected and harmonized our paper.
- Because in the title is vitamin D I suggest giving few sentences about problem of vitamin D difficiency in NAFLD to explain the aim of the study.
Thank you, we have introduced a paragraph, highlighting the aim of the study.
“ Vitamin D is an important hormone that presents multiple roles apart from the homeostasis of bone. Important papers in the field have presented a crucial role of vitamin D in the modulation of the immune system, presenting important anti-inflammatory and anti-fibrogenic roles, especially in liver fibrosis [14–16]. Recent meta-analysis depicts an important association between vitamin D status and the presence of MAFDL, both in adults and children [17–20].
The main objective of this study is to associate the histology results of liver biopsy in patients who underwent bariatric surgery for obesity, with the presence of diabetes and the status of vitamin D. The result is to estimate the prevalence of NASH and fibrosis and to verify if there are any correlation with the biochemical features such as vitamin D, lipids biochemical profile and the current medications.”
- The main objective of this study is to associate the histology results of liver biopsiesin patients.........(lines 71-74). In the results there is no answer to main objective of the study. Please re-edit the aim of the study.
Thank you, we have corrected this aspect accordingly.
“ The main objective of this study is to associate the histology results of liver biopsy in patients who underwent bariatric surgery for obesity, with the presence of diabetes and the status of vitamin D.”
- There is no information about the source of grading system used in determination of fibrosis grade [refferences].
Thank you, we have corrected this overview.
- Was there a difference in the grade of fibrosis between grade II obesity and morbid obesity patients?
Thank you for this wonderful question, we did search to see if we have significant difference, but there were no significant differences between grade II and morbid obesity, and unfortunatelly the lot of patients was too little to have a great impact in the statistics.
- Lines 165-169 (Literature studies............cardiovascular diseases [18]) should be placed in the discussion not in the results.
Thank you, we have corrected this and placed the paragraph in the correct section.
“Literature studies have identified a strong correlation between obesity, liver steatosis and low vitamin D levels.
There are numerous literature studies determining the association between vitamin D and MAFLD. Targher et al [31] conducted the first to study proving the association between MAFLD and vitamin D levels, confirming that vitamin D levels were lower in patients with MAFLD compared to controls. Furthermore, vitamin D serum concentra-tions can predict the histological severity of MAFLD, NASH patients having lower vitamin D levels compared to the levels found in patients with isolated fatty liver disease, proven by many other literature studies [32–35]. Knowing that vitamin D undergoes an important biological activation in the liver, it is biologically possible that chronic liver diseases, including MAFLD, could alter the vitamin D metabolism, decreasing its plasmatic levels [36]. However, further research is necessary.
It was observed in many studies including meta-analysis that vitamin D concen-trations are linked to a decrease of the risk of occurrence of metabolic syndrome, diabetes, and cardiovascular diseases [28–30].”
- Were there any patients in the study group with very low vitamin D concentrations, e.g. <10 or 14 ng / mL, and did they have a higher degree of fibrosis or steatosis? If possible, it would be worth checking. You mentioned about such situation in the discussion.
Thank for this interesting question. We did have several patients with a vitamin D concentration bellow 10, but the distribution is low and very uneven in our cohort to draw a pertinent conclusion. This could a very interesting subject to further evaluate on a higher cohort of patients.
We hope that the changes are according to your recommendations.
Thank you,
The authors
Reviewer 3 Report
In this relatively small retrospective study the authors analyzed the risk of steatohepatitis among 49 patients who underwent bariatric surgery for obesity. Authors find the association between the presence of diabetes and steatohepatitis in this cohort of patients. Additionally, prominent finding is the presence of hypovitaminosis D in the diabetic group.
This is a retrospective study so its limitations are inherent to the nature of the study. Authors acknowledge this in limitation section. Strong feature of this study is that actually all patients were diagnosed with liver biopsy.
I have the following comments and suggestions for the authors:
1. Please consider using term MAFLD instead of NAFLD
2. Abstract is not clear- vitamin D was not mentioned at all in the goals of the study despite it being contained in the title. This should be corrected
3. Consider using MAFLD ( metabolic associated fatty liver disease) instead of NAFLD. If you disagree please provide rebuttal
4. Diverticulosis was recently found in one retrospective study to be associated with MAFLD/NAFLD and obesity. Please add this on the line 50 in introduction ( https://www.ncbi.nlm.nih.gov/pmc/articles/PMC8778461/)
5. Results section- the first paragraph should report how many patients were diabetic
6. Table 1- baseline HgbA1c was only 6.1- I am confused if this is total 49 patients or this is just diabetic patients? It would be better to report this information more clearly
7. Line 166- instead of vitamin D level - it would be better to add low vitamin D levels
8. Discussion - line 200: it should be added that vitamin D deficiency is one of the reason why people with NAFLD are at higher risk for both viral and bacterial infections. Please see the following : https://www.mdpi.com/2072-6643/9/9/1015/htm and https://pubmed.ncbi.nlm.nih.gov/33977096/
9. references need to be expended and newer ones used. Some of the references are more than 10 years old despite the fact that the research on this topic is relatively new and quite a bit has been published within the last 5 years. Some references are even from 2003
Author Response
Honorable reviewer,
Thank you for your time and review suggestions, we found them very helpful. We have made the changes according to your suggestions, as it follow:
- Please consider using term MAFLD instead of NAFLD
Thank you, we have changed our terminology accordingly.
- Abstract is not clear- vitamin D was not mentioned at all in the goals of the study despite it being contained in the title. This should be corrected
Thank you for your suggestion, we have corrected this point and included it in primary endopoints of the study.
“The primary endpoint of the study was to study the relationships of the steatohepatitis/liver fibrosis with the presence of diabetes and antidiabetic drugs and to evaluate if there are any current relationships between the biochemical workups. A special importance was accorded to the correlations between vitamin D levels and the presence of hepatic steatosis.”
- Consider using MAFLD ( metabolic associated fatty liver disease) instead of NAFLD. If you disagree please provide rebuttal.
Thank you, we have changed our terminology accordingly.
- Diverticulosis was recently found in one retrospective study to be associated with MAFLD/NAFLD and obesity. Please add this on the line 50 in introduction ( https://www.ncbi.nlm.nih.gov/pmc/articles/PMC8778461/)
Thank you, we have included the retrospective and it’s important findings, as it follows.
“It can present different degrees of histological modifications, from simple hepatic steatosis and nonalcoholic steatohepatitis (NASH), presenting an inflammatory reaction, cellular damage in the liver such as ballooning of hepatic cells, presenting with or without different degrees of fibrosis, progressing to cirrhosis and eventually to hepatocellular carcinoma MAFLD is commonly associated with obesity, type 2 diabetes (T2DM), dyslipidemia, and insulin resistance, all components of the metabolic syndrome, the presence of metabolic dysregulation and colonic diverticulosis [5] , implying that MAFLD is the hepatic involvement of the syndrome [6]”.
- Results section- the first paragraph should report how many patients were diabetic
Thank you for your suggestion, we have added the missing information.
“ Our study included 49 patients, 18 diabetic ( 2 patients with type 1 diabetes and 16 patients with type 2 diabetes mellitus) and 31 non-diabetic patients who underwent bariatric surgery by the same operating team. Of those patients, 14 (28,6%) had grade II obesity and 35 (71,4%) had morbid obesity. From the cohort, 34 patients were women and 15 men, with ages between 29 and 63 years, median age was 50 years.”
- Table 1- baseline HgbA1c was only 6.1- I am confused if this is total 49 patients or this is just diabetic patients? It would be better to report this information more clearly
Thank you for the suggestion, we did not clearly note the information The HbA1C results presented in Table 1 are from diabetic and non-diabetic patients, as all of them were evaluated with this parameter before surgery.
“Baseline characteristic of the studied cohort are presented in Table 1. The HbA1C results presented in Table 1 are from diabetic and non-diabetic patients, as all of them were evaluated with this parameter before surgery.”
- Line 166- instead of vitamin D level - it would be better to add low vitamin D levels
Thank you, we have made the corrections.
“Literature studies have identified a strong correlation between obesity, liver steatosis and low vitamin D levels. D levels.”
- Discussion - line 200: it should be added that vitamin D deficiency is one of the reason why people with NAFLD are at higher risk for both viral and bacterial infections. Please see the following : https://www.mdpi.com/2072-6643/9/9/1015/htm and https://pubmed.ncbi.nlm.nih.gov/33977096/
Thank you for your recommendation. We have introduced the informations in our study.
“Furthermore, strong evidence from the literature database show that MAFLD can presents serious extrahepatic manifestations such as cardiovascular disease, polycystic ovarian syndrome, chronic kidney disease, and hypothyroidism, as well as a higher risk of developing various infections [28,29].”
- References need to be expended and newer ones used. Some of the references are more than 10 years old despite the fact that the research on this topic is relatively new and quite a bit has been published within the last 5 years. Some references are even from 2003
Thank you, we have added new and significant papers to our references.
We hope that the changes are according to your recommendations.
Thank you,
The authors
Round 2
Reviewer 1 Report
1. Despite several modifications, authors have to go for another round of language correction because the errors were apparent from the beginning of the manuscript
2. Table 1: Legends - LDL and HDL should be lipoprotein
Author Response
Honorable reviewer,
Thank you for your time and review suggestions, we found them very helpful. We have made the changes according to your suggestions, as it follow:
- Despite several modifications, authors have to go for another round of language correction because the errors were apparent from the beginning of the manuscript
Thank you, we are currently working on language corrections, as we are not native English speakers.
- Table 1: Legends - LDL and HDL should be lipoprotein
Thank you, we have made the changes.
We hope that the changes are according to your recommendations.
Thank you,
The authors
